# Oral Transmucosal Cannabidiol Oil Formulation as Part of a Multimodal Analgesic Regimen: Effects on Pain Relief and Quality of Life Improvement in Dogs Affected by Spontaneous Osteoarthritis

**DOI:** 10.3390/ani10091505

**Published:** 2020-08-26

**Authors:** Federica Alessandra Brioschi, Federica Di Cesare, Daniela Gioeni, Vanessa Rabbogliatti, Francesco Ferrari, Elisa Silvia D’Urso, Martina Amari, Giuliano Ravasio

**Affiliations:** 1Department of Veterinary Medicine, Università degli Studi di Milano, 20122 Milan, Italy; federica.brioschi@unimi.it (F.A.B.); daniela.gioeni@unimi.it (D.G.); 2Department of Health, Animal Science and Food Safety, Università degli Studi di Milano, 20122 Milan, Italy; federica.dicesare@unimi.it; 3Department of Veterinary Medicine, Centro Clinico Veterinario e Zootecnico Sperimentale, Università degli Studi di Milano, 20122 Milan, Italy; vanessa.rabbogliatti@unimi.it (V.R.); francescoferrari01@libero.it (F.F.); martina.amari@hotmail.it (M.A.); 4CVRS-Policlinico Veterinario Roma Sud, 00173 Rome, Italy; elisasilvia.durso@gmail.com

**Keywords:** Canine Brief Pain Inventory, cannabidiol, chronic pain, dogs, oral transmucosal administration, osteoarthritis, pain management

## Abstract

**Simple Summary:**

Osteoarthritis is a progressive and degenerative condition that affects dog populations, causing pain. The pain associated with osteoarthritis is considered to be chronic, owing to both active inflammation and to a maladaptive component caused by central sensitization. Chronic pain in dogs is being increasingly recognised as a significant problem, and finding successful treatments against canine osteoarthritis-related pain is challenging. The aim of this study was to assess the efficacy in pain management over a twelve-week period of oral transmucosal cannabidiol, in combination with a multimodal pharmacological protocol, in dogs affected by spontaneous osteoarthritis. Dogs receiving oral transmucosal cannabidiol in addition to an anti-inflammatory drug, gabapentin and amitriptyline showed a meaningful improvement in Canine Brief Pain Inventory scores, in comparison with dogs that did not receive cannabidiol. The present study suggests that the addition of oral transmucosal cannabidiol to a multimodal pharmacological treatment for canine osteoarthritis improves owner reported pain scores and quality of life of dogs, without severe adverse effects.

**Abstract:**

The aim of this study was to evaluate the efficacy of oral transmucosal (OTM) cannabidiol (CBD), in addition to a multimodal pharmacological treatment for chronic osteoarthritis-related pain in dogs. Twenty-one dogs were randomly divided into two groups: in group CBD (*n* = 9), OTM CBD (2 mg kg^−1^ every 12 h) was included in the therapeutic protocol (anti-inflammatory drug, gabapentin, amitriptyline), while in group C (*n* = 12), CBD was not administered. Dogs were evaluated by owners based on the Canine Brief Pain Inventory scoring system before treatment initiation (T0), and one (T1), two (T2), four (T3) and twelve (T4) weeks thereafter. Pain Severity Score was significantly lower in CBD than in C group at T1 (*p* = 0.0002), T2 (*p* = 0.0043) and T3 (*p* = 0.016). Pain Interference Score was significantly lower in CBD than in C group at T1 (*p* = 0.0002), T2 (*p* = 0.0007) and T4 (*p* = 0.004). Quality of Life Index was significantly higher in CBD group at T1 (*p* = 0.003). The addition of OTM CBD showed promising results. Further pharmacokinetics and long-term studies in larger populations are needed to encourage its inclusion into a multimodal pharmacological approach for canine osteoarthritis-related pain.

## 1. Introduction

Osteoarthritis (OA) is a progressive and degenerative condition that affects dog populations and causes pain and crepitus in joints, decreased mobility and reluctance to exercise [1]. It is one of the main causes of chronic pain in dogs, owing to both active inflammation and to a maladaptive component caused by central sensitization [2]. Management of osteoarthritic pain includes treatment with anti-inflammatory drugs; non-steroidal (NSAIDs) or corticosteroids. The potential side effects of these drugs may preclude long-term use, particularly in geriatric patients with comorbidities, such as kidney and gastrointestinal diseases [3,4]. Furthermore, clinical experience [5] and a review of experimental studies [6,7] clearly state that anti-inflammatory drugs do not provide complete pain relief in dogs with OA. Adjunctive medications with analgesic properties (e.g., gabapentin and amitriptyline) are used in combination with anti-inflammatory therapy in human patients [8], and a similar approach has been suggested in dogs [5]. Gabapentin is an anticonvulsant drug that exerts its analgesic effects via a blockade of voltage-dependent calcium channels [9]. Due to this mechanism of action, it can be used in dogs affected by OA for pain management with minimal side effects, though owners should be warned about possible sedation when beginning administration [10]. Amitriptyline is a tricyclic antidepressant drug that inhibits the reuptake of serotonin and norepinephrine in the central nervous system, and is therefore expected to reinforce the descending inhibitory nociceptive modulation [9]. To the authors’ knowledge, there are no clinical trials or experimental studies evaluating the use of amitriptyline for OA-related pain in dogs, but an insight for their use can be gathered in human literature [11]. Anti-inflammatory drugs, gabapentin and amitriptyline are available options for long-term treatment in osteoarthritic dogs that experienced states of chronic unmanaged pain. However, there is still a lack of knowledge regarding their efficacy, whether administered alone or in combination [2]. Moreover, since the lack of consensus in canine OA-related pain management, there is a constant search to find alternative therapies, and new treatments are often suggested and embraced, despite the lack of proven clinical effectiveness [12].

Over the last three decades, a new biochemical and physiological receptor system, the endocannabinoid system, has been described [13]. The endocannabinoid receptor system, composed of two cannabinoid receptors (CB1 and CB2) and their ligands, plays a role in pain modulation and inflammation attenuation [14]. Cannabinoid receptors are widely distributed throughout the central and peripheral nervous system [13], and are also present in the human synovium [15]. Cannabidiol (CBD) is a non-psychotropic cannabinoid that exerts immunomodulatory, antihyperalgesic, antinociceptive and anti-inflammatory effects, acting as a non-competitive allosteric antagonist of CB receptors [16]. Given these pharmacological properties, CBD represents an attractive therapeutic option in dogs with OA [17]. Unfortunately, its bioavailability has been reported to be extremely low when given orally to dogs, presumably due to high first-pass effect through the liver [18].

Oral transmucosal (OTM) route is gaining importance in veterinary medicine, because of the advantages it offers over oral, intramuscular and intravenous administration for systemic drug delivery [19,20,21]. These major advantages are its easy practicability, lack of pain during administration, high blood mucosal supply and avoidance of the hepatic first-pass effect or gastrointestinal degradation [22,23].

The purpose of the study was to evaluate the efficacy of a CBD oil formulation, included within a multimodal pharmacological regimen, in alleviating pain in dogs affected by spontaneous OA, following OTM administration. The secondary objective included the identification of any adverse clinical effect associated with 12-week multimodal pharmacological therapy, and in particular with CBD oil administered through the OTM route. The authors hypothesized that CBD oil, administered to the buccal mucosa of dogs, would enhance the effectiveness of a selected multimodal analgesic protocol for the treatment of OA-related pain, without causing greater side effects.

## 2. Materials and Methods

### 2.1. Animals

This study was approved by the Institutional Ethical Committee for Animal Care at the University of Milan (OPBA_15_2020), and all dogs were enrolled for CBD oil administration, after obtaining owner’s written informed consent. The study included twenty-four client-owned dogs, of different breed, age, body weight and gender, presented to the Veterinary Teaching Hospital of the University of Milan (Lodi, Italy), for evaluation and treatment of pain related to OA. Inclusion criteria were: radiographic evidence of OA (i.e., periarticular osteophytes, irregular or narrowed joint space and subchondral bone sclerosis), OA-associated signs of joint dysfunction (i.e., lameness, difficulty lying down, standing up, going up or down stairs, reluctance to jump, or difficulty jumping) and painful joint(s) on palpation. Radiographic findings and OA localization were noted and recorded by a boarded radiologist. Patient screening at baseline (T0) included a physical examination, blood cell count and serum biochemical analysis. Exclusion criteria included demonstrated neurologic, neoplastic, renal or uncontrolled endocrine disease and history of coagulopathy. Dogs that received anti-inflammatory medications and/or other analgesic therapies or that underwent orthopedic procedures within four weeks prior to the initial evaluation were excluded from the study.

### 2.2. Study Design

Dogs were enrolled over a period of 12 months, and were involved in a 12-week multimodal therapeutic program for OA-related pain treatment. Upon enrollment, all subjects were randomly assigned to two groups (CBD and C), using a commercial software program (Microsoft Office Excel 2013; Microsoft Corp, Redmond, WA, USA). Regardless of the group considered, all dogs were orally administered an anti-inflammatory drug (i.e., firocoxib or prednisone), gabapentin and amitriptyline. In CBD group, patients received also a CBD oil at the dose of 2 mg kg^−1^ every 12 h, which was added to the multimodal pharmacological protocol, and was administered by OTM route. In C group, the administration of CBD was not included. Firocoxib was the first-choice anti-inflammatory treatment. In the case of reported adverse effects following NSAIDs assumption, prednisone administration was decided as an alternative to firocoxib. Specifically, the dose of anti-inflammatory medications was lowered during the observational study period, as follows: treatment was given orally for the first week at a standard dose (5 mg kg^−1^ every 24 h for firocoxib, or 0.5 mg kg^−1^ every 12 h for prednisone), then, the daily dose was reduced by 50% during the second week and decreased again by 50% during the remaining study period. In the case of poor response to dosage lowering, defined as an increase of ≥1 in Pain Severity Score (PSS) and/or ≥2 in Pain Interference Score (PIS) [24], the anti-inflammatory daily dose was restored to the previous higher dosage. Dogs also received oral gabapentin (10 mg kg^−1^ every 12 h during the first week, 5 mg kg^−1^ every 12 h during the remaining study period) and oral amitriptyline (1 mg kg^−1^ every 24 h for the entire study period).

The CBD oil used in this study was a galenic formulation that can be prepared and sold only in authorized pharmacies. The CBD oil contained 40, 100, or 200 mg of CBD mL^−1^, according to the patient weight, with only trace amounts of the other cannabinoids (<0.01 mg mL^−1^). The remaining ingredient was medium chain triglycerides (MCT) oil. Access to food was withheld for one hour before CBD administration, and was reinstated one hour post treatment. Water was given ad libitum. Oral transmucosal administration of the CBD oil was performed by the owner using a syringe without a needle, inserted into the buccal pouch.

To assess the dog’s pain and quality of life, owners were contacted via email and asked to complete the Canine Brief Pain Inventory (CBPI), a validated numeric rating scale-based questionnaire, which contained 11 questions on the dog’s lameness, mood and willingness to move, play and jump [25,26]. Four questions required the owners to grade the severity of their dog’s pain over the previous days. The 4 pain severity questions were scored on a discrete numerical scale of 0 (no pain) to 10 (extreme pain); the responses for these questions were averaged to generate the PSS [25]. Six questions evaluated the pain interference with dog’s general activity, enjoyment of life and locomotive function. The 6 pain interference questions were scored on a discrete numerical scale of 0 (does not interfere) to 10 (completely interferes); the responses for these questions were averaged to generate the PIS [25]. In addition, a final question was included at the end of the questionnaire, to obtain the owner’s overall assessment of the dog’s quality of life (Quality of Life Index, QoL) [26]. Question 11 (QoL) was graded on a discrete 0 to 4 numerical scale, with 0 representing a poor quality of life and 4 an excellent quality of life. The owners received an Italian version of the CBPI questionnaire, translated and reviewed by three authors, who were expert in chronic pain management and fluent in the original and target languages. All of the owners were asked to evaluate their dogs based on CBPI scoring system before treatment initiation (T0), and at one (T1), two (T2), four (T3) and twelve weeks (T4) thereafter. Mean CBPI results for each time point were compared between CBD and C groups, and mean CBPI results for T1, T2, T3 and T4 were compared with T0 within each group. Individual treatment success, defined as a reduction of ≥1 in PSS and ≥2 in PIS [24], was also calculated. Furthermore, owners were asked to record the occurrence of any mild to moderate or severe adverse event; mild ptyalism and temporary somnolence were considered mild to moderate adverse effects (slightly interfering with the dog’s usual habit), while serious ptyalism, gastrointestinal disorders, lethargy and changes in behavior or distress were considered severe (significantly interfering with the dog’s usual habit). Blood cell count and serum biochemical analysis were performed at the end of the twelve-week evaluation period.

### 2.3. Statistical Analysis

An a priori sample size calculation was performed to determine the number of dogs needed for this study, with 80% power, an alpha level of 0.05 and a 95% confidence interval, using prior data, suggesting a Canine Brief Pain Inventory (CBPI) total score change of 3 out of 20 points from baseline as an indicator of successful treatment, with a standard deviation of 4 out of 20 points [27]. Calculations assessed that 7 dogs for each group would be necessary to find differences in outcomes of interest. Statistical analysis was performed using PASW 18.0 (SPSS Inc, Chicago, IL, USA). The assumption of data normality was examined by a Shapiro–Wilk test with an α = 0.05 level. Results are presented as mean ± standard deviation (SD) or as number of patients (%), where appropriate. For continuous variables that were normally distributed, comparisons between CBD and C groups were performed with independent Student’s *t*-test. The same approach was used to assess differences for each group in relation to time. For categorical variables, Fisher exact test was used to compare differences between the treatment groups. Statistical significance was set at *p* < 0.05.

## 3. Results

Twenty-one out of 24 client-owned dogs met the inclusion criteria, were enrolled in the study, and assigned to the CBD group (*n* = 9) or to the C group (*n* = 12). Reasons for withdrawal for the other three dogs included the presence of neurologic abnormalities during baseline evaluation (*n* = 1 dog in the C group) and owner’s inability to return CBPI questionnaire (*n* = 2 dogs in the CBD group). Table 1 summarizes dogs’ information about radiographic findings and OA localization of recruited dogs. Table 2 summarizes dogs’ information about breed, age, weight, gender and dosages of the analgesics included in the multimodal protocol. The statistical analysis detected no differences between group CBD and C, with respect to age (*p* = 0.07), weight (*p* = 0.06) and gender (*p* = 1.00), highlighting the homogeneity of groups.

Baseline scores for PSS (5 ± 2 in CBD group and 6 ± 2 in C group, *p* = 0.29), PIS (6 ± 2 in CBD group and 7 ± 2 in C group, *p* = 0.24) and QoL (3 ± 1 in CBD group and 2 ± 1 in C group, *p* = 0.12) were similar between groups. Pain Severity Score was significantly lower in CBD than in C group at one (T1), two (T2) and four weeks (T3) after treatment initiation: 3 ± 2 versus 7 ± 2 (*p* = 0.0002), 3 ± 1 versus 5 ± 2 (*p* = 0.0043) and 3 ± 2 versus 5 ± 2 (*p =* 0.016), respectively. Pain Interference Score was significantly lower in CBD than in C group at one (T1), two (T2) and twelve weeks (T4) after treatment initiation: 2 ± 1 versus 7 ± 2 (*p* = 0.0002), 3 ± 1 versus 6 ± 2 (*p* = 0.0007) and 2 ± 1 versus 6 ± 2 (*p* = 0.004), respectively. Quality of Life Index was significantly higher in CBD than in C group at T1: 4 ± 1 versus 2 ± 1 (*p* = 0.003), respectively. The PSS, PIS and QoL scores of the dogs recruited in CBD and C group are reported in Table 3.

Within CBD group, the comparison of mean PSS, PIS and QoL scores between T0 and each successive time point showed a decrease in PSS between baseline and T2 (*p* = 0.01) and between baseline and T3 (*p* = 0.03). Pain Interference Score (PIS) was significantly lower in CBD group at T1 (*p* = 0.001), T2 (*p* = 0.0007), T3 (*p* = 0.04), and T4 (*p* = 0.004) compared to baseline. In CBD group, QoL increased at T1 (*p* = 0.008), T2 (*p* = 0.04), and T4 (*p* = 0.01) compared with baseline. Despite no significant variations in PSS, PIS and QoL scores between baseline and other examined periods, dogs assigned to group C experienced a decrease in pain scores and an improvement in QoL.

Treatment was successful in reducing PSS in 6 out of 9 (67%) dogs of group CBD at T1, T2 and T3, and in 5 out of 9 dogs at T4 (56%). In group C, considering PSS, treatment was classified as successful in 1 out of 12 (8%) dogs at T1, 2 out of 12 (17%) dogs at T2 and T4 and 3 out of 12 (25%) dogs at T3. When considering PIS, treatment in group CBD was successful in 6 out of 9 (67%) dogs at T1 and T2, 5 out of 9 (56%) dogs at T3 and 4 out of 9 (44%) dogs at T4. In group C, considering PIS, treatment was classified as successful only in one dog (8%) at T2, T3 and T4.

Within group CBD, 7 out of 9 (78%) dogs received firocoxib, and 2 out of 9 (22%) received prednisone. Within group C, 9 out of 12 (75%) dogs received firocoxib and 3 out of 12 (25%) received prednisone. No statistical differences between CBD and C groups were observed for firocoxib (*p* = 0.47) and prednisone (*p* = 0.47) administration. In addition, in 2 out of 7 (29%) dogs (CBD group) and 4 out of 9 (44%) dogs (C group) OA-related symptoms worsened shortly after firocoxib therapy was reduced to the lowest dose. However, increasing firocoxib to 50% of the standard dose resulted in reversal of this worsening.

In all dogs, oral transmucosal CBD administration was well tolerated, with mild or absent gastrointestinal side effects. In two dogs in CBD group (2 out of 9, 22%), minimal ptyalism was observed, while in one dog in CBD group and in two dogs in C group (3 out of 21, 14%), somnolence and mild ataxia were reported. No relevant changes in the measured blood cell count and serum biochemical analysis were noted in either the CBD or C groups at the end of the twelve-week evaluation period (data not shown).

## 4. Discussion

To the authors’ knowledge, the present study is the first to evaluate the clinical effects of the OTM administration of CBD oil in dogs. A significant reduction in perceiving pain and a significant increase in quality of life was achieved in dogs affected by spontaneous OA receiving OTM CBD oil (2 mg kg^−1^ every 12 h), in addition to a multimodal analgesic regimen, compared with findings in dogs of the control group.

Because of the complex neurobiology of chronic pain, it is reasonable to believe that multimodal pharmacologic therapy is advantageous for the treatment of OA [28], although this approach has received poor attention in the veterinary literature [29]. Furthermore, the use of a multimodal therapeutic approach may reduce doses of analgesics and therefore their adverse effects [30]. The present study included a wide range of analgesic drugs, strengthening the importance of a multimodal treatment in dogs with osteoarthritic chronic pain. Osteoarthritis can cause hyperalgesia and evolve into neuropathic pain [31], therefore, the use of analgesic adjuvants, such as amitriptyline and gabapentin, appears advisable. Despite the lack of high-quality evidence to support their use, in the authors’ experience, gabapentin and amitriptyline have provided the most interesting results in pain relief in addition to NSAIDs therapy in dogs with OA. At present, NSAIDs and glucocorticoids are the most widely used drugs for OA treatment in animals [32]. The effects of these two groups of pharmaceuticals are similar, as they both have anti-inflammatory effects, have direct effects on cartilage metabolism and may stimulate the synthesis of interleukin-1 [33,34]. The number of dogs that received NSAIDs or glucocorticoids in this study was similar between groups. Thanks to the similarity between the two treatment groups and to the proved affinity between the effects on OA of NSAIDs and glucocorticoids, it is possible to make a comparison between CBD and C groups.

In the present study, the CBPI questionnaire was used to detect changes in pain scores, and to identify differences in terms of pain relief and quality of life improvement in response to treatment. This scoring system was specifically designed to quantify the intensity of pain and its impact on daily activities in dogs in their environment, and it has been validated as an owner tool to assess OA-related pain [24,25]. The questionnaire is divided into a PSS, that assesses the magnitude of pain of an animal, a PIS, that assesses the degree by which pain affects daily activities and a global assessment of the quality of life [25,26]. In the present study, the increase in comfort and activity for dogs included in the CBD group was represented by lower PSS and PIS mean values, as well as higher QoL mean values, compared with group C, at each time point. Although these values did not always differ significantly, the improvement in PSS, PIS and QoL scores was consistent. In fact, changes from baseline values were found to be significantly different in group CBD at T2 and T3 for PSS, and at every time point for PIS. In CBD group, QoL increased at T1, T2, and T4 compared with baseline. Despite no significant variations in PSS, PIS and QoL scores between baseline and other examined periods, dogs assigned to group C experienced a progressive decrease in pain scores and an improvement in QoL. These findings, although not statistically significant, allow authors to suppose that even the combination between an anti-inflammatory, gabapentin and amitriptyline resulted in some beneficial effects in terms of pain relief and quality of life improvement. It is also possible that significant results within C group could be observed with a larger sample size.

Recent evaluation of the ability of the CBPI to detect a significant improvement in osteoarthritic dogs treated with carprofen found that a decrease in PSS ≥ 1 and a decrease in PIS ≥ 2 resulted in the highest statistical power to predict whether a treatment would lead to a response in an individual dog [24]. When considering individual results in the present study, treatment success was obtained in more dogs in CBD group, compared with C group. Our results suggest that the changes detected might be due to a positive response to CBD OTM treatment, also in long-term use. In fact, a significative improvement in the CBPI scores was shown also at T4, after 12-week treatment with CBD. The use of oral CBD oil for osteoarthritic pain management in dogs has been previously studied [17]. This study demonstrated a significant decrease in PSS and PIS and a significant increase in dogs’ activity at week 2 and 4, when compared to baseline, but long-term efficacy was not evaluated [17].

Pharmacologically, CBD has a complex signalling mechanism. It can both activate and silence cannabinoid receptors, as well as modulate cannabinoid receptor pathways, influencing nociceptive signalling and reducing long-term inflammation progression [35]. Including CBD in a multimodal drug treatment is a strategy that the authors have used in order to manage more effectively OA-related pain in dogs. The results showed that this approach can be effective, suggesting that CBD may enhance concurrent analgesic drugs effects, probably by exerting a positive modulation at glycine and vanilloid TRPV1 receptors, which play a central role in the development of OA [35].

Moreover, it is well known that cannabinoid system could be exposed to degradation by cyclooxygenase type 2 (COX-2), and that this important degradative pathway might convert cannabinoids into pro-inflammatory and pro-nociceptive mediators, such as prostamides, prostaglandins and prostacyclin glycerol esters [36]. Consequently, NSAIDs that inhibit COX-2 could attenuate cannabinoids breakdown, prolonging its effects, and selectively prevent the formation of pro-inflammatory and pro-nociceptive mediators [37]. Furthermore, it has been demonstrated that COX-2 plays a role in central sensitisation, and that COX-2 inhibitors can prevent this process [38]. Authors strongly advise the use of NSAIDs that inhibit COX-2 (unless specifically contraindicated) as part of a multimodal treatment for osteoarthritic pain in dogs, especially if CBD is co-administered, since this pharmacological interaction could lead to a progressive reduction in pain perceived by the animal. The benefits of this long-term therapy could include the better control of pain, greater improvements in mobility and the potential slowing down of the osteoarthritic process through improved joint usage, even if the continuous administration of anti-inflammatories might lead to an increased incidence of adverse events. As a result, recent human guidelines suggest the administration of the lowest effective dose of NSAID to minimize side effects, and only for the time required [39]. To date, there are no studies concerning the long-term use of anti-inflammatory drugs in dogs, while a study conducted in human medicine by Luyten and colleagues (2007) [40] showed that there were no significant differences between patients exposed to either long-term or intermittent NSAIDs treatment, except for the intake of rescue analgesia, which was less frequent in the long-term treatment group. Another study, by Gunew and colleagues (2008) [41], reported that oral meloxicam was safe for long-term treatment of OA in cats, including those of advanced age. In the present study, firocoxib or prednisone were gradually decreased over time, in order to reach the lowest effective dose, as a possible solution for the challenge of long-term osteoarthritic pain treatment in dogs. Concurrent analgesic therapies may have helped to reduce anti-inflammatory effective dosage, especially in dogs that received CBD in addition to the multimodal pharmacological protocol; in fact, the subjects in CBD group experienced a better response to firocoxib reduction to the lowest dose, in comparison to dogs assigned to C group. Thus, according to the authors, including CBD and concurrent analgesic therapies (i.e., gabapentin and amitriptyline) within a multimodal analgesic protocol, seems to be a promising strategy in dogs affected by OA, in order to minimize adverse effects occurrence associated to long-term anti-inflammatory drugs consumption.

A recent study in dogs has shown that the delivery of CBD through an oil base appears to be the preferential method for absorption, while oil beadlets and transdermal do not appear as effective as infused oils [42]. In fact, the oil-based vehicle seems to be the first choice, due to the lipophilic nature of CBD [43]. Unfortunately, CBD bioavailability has been reported to be low (ranged from 13 to 19%) when given orally to both dogs and humans, presumably due to high first-pass effect from the liver [18,19,20,21,22,23,24,25,26,27,28,29,30,31,32,33,34,35,36,37,38,39,40,41,42,43,44], together with its demonstrated poor gastrointestinal permeability [45]. As the drug has low aqueous solubility and undergoes first-pass metabolism, alternative delivery routes are needed to achieve successful therapeutic effects by bypassing the first-pass effect. To confirm this statement, unpublished authors’ data pointed out inadequate pain relief following oral CBD administration, as a part of multimodal therapeutic protocol in dogs affected by spontaneous OA. In the aforementioned patients, the administration of CBD oil via OTM route, instead of oral route, resulted in satisfactory pain relief, and in quality of life improvement. OTM route allows one to avoid the first-pass metabolic effect and gastrointestinal degradation observed for the orally administered drugs. The rich blood supply of the oral mucosa allows drugs administered by this route to reach systemic therapeutic concentrations [22,23]. Moreover, OTM route represents an attractive alternative to other drug delivery routes, being a non-invasive, pain-free technique, which requires minimal restraint and does not cause distress in patients [46]. The easy practicability for the owner is another major advantage, requiring minimal technical skills compared to other routes of administration [23]. In humans, the development of an oromucosal spray that contains a roughly 1:1 ratio of THC and CBD has provided a non-invasive method of administration, that has proven to show clinically significant improvements for the symptomatic relief of chronic uncontrolled pain in advanced cancer patients [47]. However, the OTM route can be more variable than IV or IM administration due to the possibility of swallowing the delivered dose, loss of the drug outside of the mouth, expelled medication by coughing or spitting and vomiting, or ptyalism reducing or diluting the quantity of drug for absorption [48]. The oil formulation of CBD administered in the present study was flavourless, and this aspect may have prevented the incidence of marked ptyalism and vomiting. Mild and transient ptyalism was observed in two out of 9 dogs receiving CBD, while vomiting was absent, suggesting a suitable drug formulation palatability. Moreover, the oil formulation contained 40, 100, or 200 mg of CBD mL^−1^, based on the patient weight, in order to minimize the administered volume. In fact, smaller OTM volumes result in being clinically more effective, having less chance of inducing swallowing, ptyalism and/or loss of drugs outside the mouth [49]. Further studies, including a pharmacokinetic investigation of oral transmucosal administration of CBD, alone or in combination with other pharmacologic therapies, are required, in order to assess the bioavailability of this drug administered by this type of route in dogs. Somnolence and mild ataxia were observed in one dog in CBD group and in two dogs in C group, but these adverse effects were transient and resolved immediately after gabapentin dosage reduction to 5 mg kg^−1^ every 12 h. Overall, there were no moderate/severe clinical adverse effects, and there was reliable pain relief and quality of life improvement.

The present study has several limitations. The effect on blood cell count and serum chemistry analysis of the 12-week treatment period was not statistically evaluated. No relevant change was noted in either the CBD or C groups, in accordance with the findings of a previous study, showing no clinically significant alterations in blood cell count and serum chemistry during 12-week CBD-rich hemp products administration in healthy dogs [50]. However, a clinical population of osteoarthritic dogs that received oral CBD oil treatment exhibited a significant increase over time in alkaline phosphatase (ALP), from baseline to week 4 [17]. Therefore, it could be prudent to monitor liver enzyme values (especially ALP) in dogs receiving CBD oil for long periods, until controlled long-term safety studies become available. Moreover, the owner compliance to the treatment evaluation may have partially affected the results of the comparison between groups. However, although the CBPI has a subjective component, studies in dogs have indicated that owners are able to assess their pet response to analgesic therapy, and that veterinarian chronic pain assessments are not as sensitive as owner assessments [51]. Although, in the present study, the English version of CBPI questionnaire was translated and reviewed by three authors who were expert in chronic pain management and fluent in the original and target languages, the Italian version of the CBPI questionnaire has not been previously validated, and further validation studies are needed. Another limitation is that a placebo oil was not administered in addition to the multimodal pharmacological protocol assigned to group C. This may have caused a placebo effect for the owners administering a specially formulated oil-medication to dogs in CBD group. However, the authors attempted to limit the potential for bias by blinding the owners of the existence of another treatment group, to ensure they considered each of the assigned multimodal protocol as potentially effective.

## 5. Conclusions

Overall, according to the CBPI scores assigned by the owner, a satisfactory pain and quality of life management was achieved in dogs receiving OTM CBD oil (2 mg kg^−1^ every 12 h), in addition to a multimodal pharmacological approach for the treatment of OA-related pain. Combined with an anti-inflammatory drug, gabapentin and amitriptyline, CBD appears to enhance osteoarthritic pain relief and quality of life improvement. Furthermore, its co-administration results in being useful in reducing the other administered drugs’ dosage, minimizing the severity and incidence of associated side effects. The high CBD patient tolerability, the easy practicability and the paucity of adverse effects of OTM route of administration may represent potential benefits for long-term therapy.

## Figures and Tables

**Table 1 animals-10-01505-t001:** Radiographic findings and osteoarthritis (OA) involved joint of the dogs recruited in cannabidiol (CBD) (*n* = 9) and C (*n* = 12) groups.

	Group	Radiographic Findings	OA Localization
1	CBD	Moderate left stifle OA with intracapsular swelling	Left stifle
2	CBD	Moderate bilateral elbow OA, mild bilateral coxofemoral OA	Left elbow, right elbow, bilateral hip
3	CBD	Severe right elbow OA	Right elbow
4	CBD	Severe medial coronoid remodelling (with fragmentation on the right) and bilateral elbow OA	Left elbow, right elbow
5	CBD	Severe right medial coronoid remodeling, and bilateral elbow OA	Left elbow, right elbow
6	CBD	Moderate left medial coronoid remodeling, severe left elbow OA	Left elbow
7	CBD	Severe right stifle OA with moderate intracapsular swelling, bilateral moderate coxofemoral OA	Right stifle, bilateral hip
8	CBD	Bilateral severe stifle OA due to cranial cruciate ligament disease	Left stifle, right stifle
9	CBD	Moderate-to-severe bilateral coxofemoral OA	Bilateral hip
1	C	Moderate right coxofemoral OA, severe left coxofemoral OA	Bilateral hip
2	C	Severe right shoulder OA, moderate right elbow OA	Right shoulder, right elbow
3	C	Severe bilateral elbow OA, moderate bilateral coxofemoral OA	Left elbow, right elbow, bilateral hip
4	C	Moderate right shoulder OA	Right shoulder
5	C	Severe bilateral elbow OA, moderate bilateral coxofemoral OA	Left elbow, right elbow, bilateral hip
6	C	Bilateral severe coxofemoral OA	Bilateral hip
7	C	Severe right elbow OA	Right elbow
8	C	Severe bilateral coxofemoral OA	Bilateral hip
9	C	Severe right elbow OA, mild left stifle OA	Right elbow, left stifle
10	C	Moderate bilateral coxofemoral OA	Bilateral hip
11	C	Moderate right shoulder OA, severe bilateral elbow OA	Left elbow, right elbow, right shoulder
12	C	Severe bilateral coxofemoral OA	Bilateral hip

**Table 2 animals-10-01505-t002:** Breed, age, weight, gender and analgesic therapies administered to the dogs recruited in CBD (*n* = 9) and C (*n* = 12) groups. SID, once daily; BID, twice daily.

	Group	Breed	Age	Weight	Gender	NSAIDs	Glucocorticoids	Gabapentin	Amitriptyline	CBD
			(months)	(kg)						
1	CBD	Mongrel	156	23	Female	Firocoxib (5–1.25 mg kg^−1^ SID)	None	10–5 mg kg^−1^ BID	1 mg kg^−1^ SID	2 mg kg^−1^ BID
2	CBD	Épagneul Breton	144	18	Female	None	Prednisone (0.5–0.12 mg kg^−1^ BID)	10–5 mg kg^−1^ BID	1 mg kg^−1^ SID	2 mg kg^−1^ BID
3	CBD	English Bulldog	96	25	Male	Firocoxib (5–2.5 mg kg^−1^ SID)	None	10–5 mg kg^−1^ BID	1 mg kg^−1^ SID	2 mg kg^−1^ BID
4	CBD	Cane Corso	125	45	Female	Firocoxib (5–2.5 mg kg^−1^ SID)	None	10–5 mg kg^−1^ BID	1 mg kg^−1^ SID	2 mg kg^−1^ BID
5	CBD	Labrador Retriever	110	45	Male	Firocoxib (5–1.25 mg kg^−1^ SID)	None	10–5 mg kg^−1^ BID	1 mg kg^−1^ SID	2 mg kg^−1^ BID
6	CBD	Dogue de Bordeaux	84	60	Male	Firocoxib (5–1.25 mg kg^−1^ SID)	None	10–5 mg kg^−1^ BID	1 mg kg^−1^ SID	2 mg kg^−1^ BID
7	CBD	Border Collie	156	20	Male	None	Prednisone (0.5–0.12 mg kg^−1^ BID)	10–5 mg kg^−1^ BID	1 mg kg^−1^ SID	2 mg kg^−1^ BID
8	CBD	Boxer	108	33	Male	Firocoxib (5–1.25 mg kg^−1^ SID)	None	10–5 mg kg^−1^ BID	1 mg kg^−1^ SID	2 mg kg^−1^ BID
9	CBD	Boxer	108	40	Female	Firocoxib (5–1.25 mg kg^−1^ SID)	None	10–5 mg kg^−1^ BID	1 mg kg^−1^ SID	2 mg kg^−1^ BID
1	C	Australian Sheperd	156	24	Male	Firocoxib (5–1.25 mg kg^−1^ SID)	None	10–5 mg kg^−1^ BID	1 mg kg^−1^ SID	None
2	C	Labrador Retriever	152	41	Male	Firocoxib (5–1.25 mg kg^−1^ SID	None	10–5 mg kg^−1^ BID	1 mg kg^−1^ SID	None
3	C	Golden Retriever	173	29	Male	Firocoxib (5–2.5 mg kg^−1^ SID	None	10–5 mg kg^−1^ BID	1 mg kg^−1^ SID	None
4	C	Cocker Spaniel	167	13	Female	Firocoxib (5–2.5 mg kg^−1^ SID	None	10–5 mg kg^−1^ BID	1 mg kg^−1^ SID	None
5	C	Labrador Retriever	161	30	Female	Firocoxib (5–1.25 mg kg^−1^ SID	None	10–5 mg kg^−1^ BID	1 mg kg^−1^ SID	None
6	C	German Sheperd	115	25	Female	Firocoxib (5–1.25 mg kg^−1^ SID)	None	10–5 mg kg^−1^ BID	1 mg kg^−1^ SID	None
7	C	Labrador Retriever	153	34	Male	None	Prednisone (0.5–0.12 mg kg^−1^ BID)	10–5 mg kg^−1^ BID	1 mg kg^−1^ SID	None
8	C	German Sheperd	108	25	Female	None	Prednisone (0.5–0.12 mg kg^−1^ BID)	10–5 mg kg^−1^ BID	1 mg kg^−1^ SID	None
9	C	Mongrel	180	10	Male	Firocoxib (5–2.5 mg kg^−1^ SID	None	10–5 mg kg^−1^ BID	1 mg kg^−1^ SID	None
10	C	Mongrel	127	22	Male	None	Prednisone (0.5–0.12 mg kg^−1^ BID)	10–5 mg kg^−1^ BID	1 mg kg^−1^ SID	None
11	C	English Bulldog	108	27	Female	Firocoxib (5–2.5 mg kg^−1^ SID)	None	10–5 mg kg^−1^ BID	1 mg kg^−1^ SID	None
12	C	Mongrel	182	18	Male	Firocoxib (5–1.25 mg kg^−1^ SID)	None	10–5 mg kg^−1^ BID	1 mg kg^−1^ SID	None

**Table 3 animals-10-01505-t003:** Pain Severity Score (PSS), Pain Interference Score (PIS) and Quality of Life Index (QoL) (adopted by Brown et al., 2008) of the dogs enrolled in CBD (*n* = 9) and C (*n* = 12) groups.

Time Point Score Sig.		T0			T1			T2			T3			T4		
		PSS a,b	PIS c,d,e,f	QoL g,h,i	PSS *	PIS *,c	QoL *,g	PSS *,a	PIS *,d	QoL h	PSS *,b	PIS e	QoL	PSS	PIS *,f	QoL i
1	CBD	4	5	3	3	2	3	2	4	3	2	4	3	2	3	4
2	CBD	2	2	3	1	1	4	1	1	4	1	1	4	2	1	3
3	CBD	5	5	3	1	3	4	3	4	3	6	6	3	6	3	3
4	CBD	9	9	1	4	5	3	3	3	3	3	4	3	3	1	4
5	CBD	5	6	2	2	1	4	3	2	4	3	3	4	5	4	4
6	CBD	9	9	2	5	3	3	5	2	2	2	6	2	3	2	2
7	CBD	5	6	3	5	4	4	4	3	3	3	4	4	3	4	4
8	CBD	3	7	3	1	1	4	3	4	3	5	6	3	5	2	3
9	CBD	6	8	3	2	2	3	3	4	3	4	5	3	4	2	4
mean		5.33	6.33	2.55	2.66	2.44	3.55	3	3	3.11	3.22	4.33	3.22	3.66	2.44	3.44
SD		2.4	2.2	0.7	1.6	1.4	0.5	1.2	1.1	0.6	1.5	1.6	0.6	1.4	1.1	0.7
1	C	4	8	3	7	7	3	4	7	4	4	4	4	3	6	4
2	C	8	8	1	8	8	1	8	9	1	8	8	2	7	8	2
3	C	9	9	2	9	8	1	8	8	2	7	7	2	7	9	3
4	C	7	8	2	8	7	2	7	7	3	7	6	3	6	7	1
5	C	4	8	2	5	8	2	4	8	2	4	6	2	3	8	2
6	C	6	6	3	9	5	3	4	5	3	6	4	3	6	5	3
7	C	5	6	3	5	5	3	3	2	4	3	4	4	2	2	4
8	C	2	3	4	3	3	4	3	3	4	2	1	4	2	3	4
9	C	8	9	1	6	9	1	7	8	2	6	7	1	7	8	3
10	C	7	9	2	7	8	1	6	8	2	7	7	2	7	9	3
11	C	7	8	2	7	7	2	7	7	2	7	6	3	6	6	3
12	C	3	5	2	5	5	2	3	5	2	3	3	2	3	5	2
mean		5.83	7.25	2.25	6.58	6.66	2.08	5.3	6.41	2.58	5.33	5.25	2.66	4.92	6.33	2.83
SD		2.2	1.9	0.8	1.8	1.7	0.9	2.0	2.2	0.9	2.0	2.1	1.0	2.1	2.3	0.9

*p* < 0.05 between groups at the same time point (*****). *p* < 0.05 intra CBD group compared to baseline (T0): PSS T0 versus PSS T2 (a); PSS T0 versus PSS T3 (b); PIS T0 versus PIS T1 (c); PIS T0 versus PIS T2 (d); PIS T0 versus PIS T3 (e); PIS T0 versus PIS T4 (f); QoL T0 versus QoL T1 (g); QoL T0 versus QoL T2 (h), QoL T0 versus QoL T4 (i). No statistical differences intra C group compared to baseline (T0).

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
