# Peer review of "Oral Transmucosal Cannabidiol Oil Formulation as Part of a Multimodal Analgesic Regimen: Effects on Pain Relief and Quality of Life Improvement in Dogs Affected by Spontaneous Osteoarthritis"

_animals, 2020, doi:10.3390/ani10091505_

Round 1

Reviewer 1 Report

Dear authors,

Thank you for this interesting study assessing the benefit of adding OTM CBD to a multimodal analgesic regime in dogs with naturally occurring OA. While I agree that the results appear very promising, the limitations of this very small, non-placebo-controlled study should be more thoroughly discussed.

SIMPLE SUMMARY & ABSTRACT

  • Line 18 and 47: Should read “populations” rather than population.

  • Line 26: I don’t believe that you can state that “The present study proved…” Such a statement requires the current findings to be replicated in a much larger, placebo-controlled study. Consider changing wording to similar to: “the present study suggests that the addition of oral transmucosal cannabidiol to a multimodal pharmacological treatment for canine osteoarthritis improves owner reported pain scores and quality of life”.

  • Lines 41 – 42. I strongly suggest that further work is required before readers are recommended to include this drug into OA treatment regimens in a clinical setting. Further clinical, long-term safety, and PK studies are required. Please modify this sentence to say that these encouraging results suggest further research is worthwhile, rather than recommending its clinical use at this stage.

INTRODUCTION

  • Lines 50 – 53: grammatical editing required, I suggest “Management of osteoarthritic pain includes treatment with anti-inflammatory drugs; non-steroidal (NSAIDs) or corticosteroids. The potential side effects of these drugs may preclude…”

  • Line 55. Omit “as a result” because the sentence goes on to describe the approach to analgesia in people with OA that are presumably not “as a result” of the canine studies mentioned in the previous sentence.

  • Are there any long-term safety studies of CBD in dogs?

  • Line 86: I would suggest that a 4-week period is not “long-term” in a study of analgesia for a chronic disease such as OA where animals would be expected to received medication for the rest of their life in many cases. Indeed, Gamble et al. refer to a 4-week period of tx as short-term in their CBD study.

METHODS

  • Dose of CBD is not described in the methods section, only in the abstract and discussion. Please add here.

  • Was a placebo oil used in group C? The placebo effect is a well-recognised and strong phenomenon. It is possible that the process of administering a specially formulated OTM medication added a placebo effect for the owners of dogs in the treatment group. If a placebo oil was not administered to dogs in group C this must be discussed as a limitation in the discussion.

  • For the study design used, power analysis should be based on detecting a difference between the groups, rather than a change in baseline. The actual difference found between the groups is far less than the 15 points used in the power analysis study, suggesting that a larger group size would be required. The group size is very small compared to other similar studies: e.g. Vijarnsorn et al. had 79 dogs in total (Vijarnsorn, M., Kwananocha, I., Kashemsant, N. et al.The effectiveness of marine based fatty acid compound (PCSO-524) and firocoxib in the treatment of canine osteoarthritis. BMC Vet Res 15, 349 (2019). https://doi.org/10.1186/s12917-019-2110-7)

  • Was the Canine BPI translated into Italian for the dog owners, and if so, has it been validated in this language? If not, this should be mentioned as a study limitation.

RESULTS

  • Data not provided for baseline PSS, PIS or QoL – just p value. The actual values are important to know. Assuming that PSS was scored out of 40, and PIS out of 60, the values obtained in both groups from T1 onwards are very low. If baseline scores were also very low this brings the clinical significance of a difference between groups into question.

  • As well as the mean +/- SD provided, a figure should be provided showing actual values for each dog in PSS, PIS, and QOL between groups at each time point.

DISCUSSION

  • Firocoxib has been shown to be an effective analgesic for canine OA in several large studies (Hanson et al., Pollmeier et al., Ryan et al.). Can authors explain why it appears not to be in the current study (i.e. low % of individual treatment success in group C)? Might this be due to low baseline CBPI scores? Inherent problems with sole reliance upon the CBPI as an outcome measure in this setting? Or was the dose of firocoxib reduced too quickly?

  • A major limitation of this study is that the benefit of CBD alone is unknown AND its benefit in addition to firocoxib at recommended doses (5mg/kg BID) is also unknown due to the rapid reduction in firocoxib dose. To my knowledge, the efficacy of the dosing regimen for firocoxib used in this study has not been well established.

Ideally, at this stage of investigation of a new drug, either a non-inferiority study or add-on superiority trial would be performed. For example, a non-inferiority study might compare a group receiving only firocoxib at a dose of 5mg/kg BID for the entire study (i.e. an established treatment method for OA in dogs) vs a group receiving CBD only (I recognise this may have led to problems with ethical approval however). An example of an add-on superiority trial would be to compare firocoxib 5mg/kg BID for the entire period (again, an established drug and dose for OA treatment) +/- CBD (without the inclusion of gabapentin and amitriptyline which lack high quality evidence to support their use). The study design cannot be changed now but this should be discussed.

  • Lack of pharmacokinetic data for the OTM route of CBD in dogs should also be discussed and is required prior to its clinical use.

  • Long-term safety of this drug has not been investigated in dogs. Again, this must be mentioned as a limitation to its clinical use.

Reviewer 2 Report

Thanks for an interesting paper. I do have a couple of questions.

What is the onset time of CBD, is it a rapid onset due to being OTM?

Gabapentin often takes days to achieve the proper dosing scheme, would you recommend starting gabapentin first with the amitriptyline and then in a few days begin the CBD? Or can they all be started at the same time?

Is gabapenin and amitriptyline given with food? you state the CBD is given one hour without food, again can all the medications be given together?

Would you recommend CBD for long term use in the dog with OA?

The goal is to decrease the dose of the other medications but keep the same dose of the CBD?

How was this supplied to the owner, a bottle and then they drew up the dose? or were the doses pre-measured?

Round 2

Reviewer 1 Report

Dear Authors, Thank you for your thorough response to my comments and questions and for your edits accordingly.